# Potential Uses of Scallop Shell Powder as a Substrate for the Cultivation of King Oyster Mushroom (*Pleurotus eryngii*)

Yuanyuan Zhou, Zihao Li, Haijun Zhang, Qingxiu Hu and Yajie Zou *

Institute of Agricultural Resources and Regional Planning, Chinese Academy of Agricultural Sciences, Beijing 100081, China; yuan12290321@163.com (Y.Z.); snowglorm@163.com (Z.L.); zhanghaijun8681@126.com (H.Z.); hqx9758@126.com (Q.H.)
* Correspondence: zouyajie@caas.cn

**Abstract:** Scallop shells are currently a major form of waste generated in the Chinese fishing industry. However, they have the potential to be used as important industrial products. This study was conducted to assess the utility of scallop shell powder (SSP) supplementation in improving the growth of king oyster mushrooms (*Pleurotus eryngii*) grown on sawdust and sugarcane bagasse substrates. The outcomes of interest included mycelial growth, yield, biological efficiency, fruiting body traits, nutrient supply, and the mineral composition of *P. eryngii*. Supplementation with SSP accelerated the mycelial growth of *P. eryngii*. The yield of fruiting bodies (399.5 g/bag) and the biological efficiency (84.6%) were 14% higher after supplementation of the substrate with 2% SSP compared with those of mushrooms grown on substrates not supplemented with SSP (349.8 g/bag and 74.0%, respectively). Moreover, the crude protein and fiber content of the fruiting bodies significantly improved after growth with SSP. Furthermore, supplementation with 2% SSP increased the calcium (Ca) content of the fruiting bodies of *P. eryngii* by 64% (to $67.2 \pm 15.7$ mg kg$^{-1}$) compared with that of mushrooms grown on a control substrate (41.0 mg kg$^{-1}$). This study revealed that *P. eryngii* can efficiently use the Ca provided by raw SSP, generating higher Ca content in their fruiting bodies. Our results demonstrate that the supplementation of substrates with SSP can be useful for enhancing both the yield and nutritional content of *P. eryngii*.

**Keywords:** *Pleurotus eryngii*; growth rate; yield; biological efficiency; nutritional value

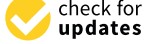



## 1. Introduction

Chinese fisheries are the largest global suppliers of shellfish. Furthermore, shellfish are the second-largest mariculture species in China. The production of scallops was estimated at 1,828,107 tons in 2019 [1], resulting in the generation of a tremendous quantity of scallop shells, which are usually abandoned in garbage dumps or discarded into the sea as solid wastes. In fact, the disposal of scallops into public waters or onto reclaimed lands has become a major environmental problem. Scanning electron micrographs (SEM) and energy-dispersive spectrometers (EDS) were used to measure the material characteristics of scallop shell power (SSP), which showed that the main content of SSP is $CaCO_3$, but it also contains 5% organic compounds by weight [2,3]. These components are valuable for many fields, including medicine, food healthcare, sewage treatment, the production of various additives, and soil treatments [4]. Studies that used SSP for the preparation of thermal insulation materials and nanoshell powders for the adsorption of pesticides have shown excellent performances for SSP [2,3].

*P. eryngii*, also called the king trumpet mushroom or king oyster mushroom, is well known for its remarkable flavor and high nutritional content, and it is cultivated and consumed worldwide [5,6]. Pharmacological studies have suggested that this mushroom contains medicinal compounds with antitumor, antiviral, and antioxidant activities [6–9]. This edible mushroom was originally cultivated in northern Italy and Switzerland, where

it is known locally as cardoncello. *P. eryngii* has continuously risen in popularity, becoming the most widely cultivated mushroom in China, Korea, and Japan over the past few decades [10]. The total yield of *P. eryngii* in China was estimated to be 2,034,512 tons in 2019 (data from the China Edible Fungi Association). More recently, with the increasing consumer demand for high-quality food, increasing the nutritional quality of this mushroom has become essential for the cultivation of *P. eryngii* in the future.

*P. eryngii* can be cultivated on a wide range of fruiting substrates including sawdust, cottonseed hulls, corn cobs, soybean meal, wheat bran, sugarcane bagasse, chopped rice straw, umbrella plant, rice husks, corn stover, wheat straw, peanut meal, korshinsk peashrub, and other agro-industrial residues [9,10]. Typically, insoluble Ca salts, such as lime and $CaSO_4$, are added to increase substrate pH (to ~7.0), reduce bacterial contamination, enhance aeration by aggregation, and improve the texture and porosity of the substrates. Several previous studies have investigated the ability of *P. eryngii* [11–13], *Pleurotus florida* [14], *Pleurotus pulmonarius* [15], *Agaricus subrufescens* [16], and *Lentinula edodes* [17] to absorb Ca from various Ca sources. It has been shown that various natural Ca sources can be used such as starfish shells for cultivation of *P. eryngii* and oyster shells for the cultivation of *P. eryngii* and *P. florida* [11–13]. These prior works studied the effect of supplementation with various amounts of Ca-containing oyster and starfish shell powders on the mycelial growth of *P. eryngii*, determining the effect of the powders on yield, spawn run, and primordial formation. Furthermore, shell powder supplementation was shown to potentially increase the Ca content in the fruiting bodies of *P. eryngii* [11,12]. Interestingly, low concentrations of oyster shell powder improved nutritional content and produced better results than high concentrations of the powder. Thus, such supplementation may be an efficient strategy for enhancing the yield and nutritional value of *P. florida* during cultivation [13]. As a result, various types of shell, including crab shells, shrimp shells, clam shells, oyster shells, scallop shells, and starfish shells, are being developed as practical additives for the industrial growth of mushrooms [11]. In line with this prior research, SSP can provide an increase in organic matter, amino acid content, and exchangeable cation concentration in the growth substrates. SSP can provide a long-term and continuous release of nutrients, help regulate pH levels, improve nutrients uptake, and enhance substrate tilt. Notably, SSP is not only a good source of Ca but also a freely available waste product.

Previously, calcined scallop shells were added to sawdust medium to test their effects on the cultivation of *P. eryngii* [12,13]. This study was performed to identify the influence of SSP that had not been calcined and to investigate SSP's ability to increase the mycelial growth rate, yield of fruiting bodies, and biological efficiency of *P. eryngii*. Furthermore, differences in the morphology, fruiting characteristics, nutritional composition, and mineral composition of the mushrooms grown on different substrates were also analyzed.

## 2. Materials and Methods

### 2.1. Inoculum Source and Spawn Preparation

*P. eryngii* (CCMSSC 003898) was obtained from the China Center for Mushroom Spawn Standards and Control, the Institute of Agricultural Resources and Regional Planning, the Chinese Academy of Agricultural Sciences. For all experiments, mycelia were grown on potato–dextrose agar (PDA) medium (potato starch 4.0 g, dextrose 20.0 g, and agar 15.0 g per liter at 121 °C for 30 min to sterilize) at 25 °C in the dark until the mycelia covered most of the plate surfaces.

### 2.2. Preparation of Ca Powder (SSP)

Scallop shells (100 kg) were collected from the Bohai Coast of China (39°13″ N, 117°2″ E). The samples were soaked in distilled water for 24 h and air-dried. The shells were mechanically broken into small pieces (10 mm × 10 mm) followed by grounding into a fine powder (particle size < 1 mm). Finally, the samples were separated with a 60 mesh separator of a vibrating sieve (Shanghai INESA Physico-Optical Instrument Co.,

Ltd., Shanghai, China). The powder was sterilized chemically with 2% (*v/v*) formalin (Solarbio, Beijing, China) before being added to the growth media.

### 2.3. Substrate Preparation

Sawdust, sugarcane bagasse, cottonseed hull, corncob, wheat bran, maize powder, soybean meal, and their combinations were used as base substrate (BS) for mushroom cultivation, and the BS was named CK. $CaCO_3$ was added to the BS as a control for a chemical-based calcium source. The weight ratio of $BS/CaCO_3$ was 99.5/0.5, and this substrate was named CK1. The SSP was precisely weighed and used to create defined combinations of BS and SSP at the following weight ratios of BS/SSP: 99/1, 98/2, 97/3, 96/4, and 96/5, which were named T1, T2, T3, T4, and T5. The dry components were completely blended, and then water was added to the mixture to create a substrate with 65% moisture [18]. The final concentration of moisture was measured by an oven-drying method in triplicate. The moist substrate (1350 g) was placed in polypropylene bags (18 cm × 38 cm); the dry weight of the mixture was 472.5 g. A single vertical hole (2 cm × 18 cm) was made at the core of the bag, which was used to inoculate the spawn. The polypropylene bags were tightened with plastic rings and vent lids and then sterilized at 121 °C for 90 min with a sterilizer (Phcbi, Osaka, Japan). The sawdust was subjected to weathering and composting in open air for half a year before it was incorporated into the substrates. All other ingredients, including SSP, were mixed into the substrate fresh without any pretreatment.

### 2.4. Assay for the Growth Rate

The growth rate of mycelia in all the substrate combinations was measured as previously described by linear growth methods [13]. Each substrate combination was individually placed in glass tubes (320 mm in length and 30 mm diameter) with a density of approximately 0.8 g cm$^{-3}$. The samples were inoculated into the top of each tube using a mycelial disk (5 mm in diameter), blocked with sterile rubber plugs, and incubated in an incubator (Phcbi, Osaka, Japan) at 25 ± 1 °C. After 7 d of incubation, the growth rate of the different treatments was calculated. The growth extension speed of mycelia was assessed by the observed spawn running times (the days for mycelia growth to fill the substrate), which was calculated over the entire growth period with no correction for the length of the lag period. This experiment was assessed in five replicates for each substrate group.

### 2.5. Mushroom Cultivation

Cultivation of *P. eryngii* was carried out following the methods described by Zhang with some modifications [19]. A stick spawn was made using polypropylene bags with dimensions of 15 cm × 30 cm × 0.04 cm. The sticks, which were of broadleaf tree wood, were 15 cm long, 0.42 cm thick, and 0.7 cm wide, with rough surfaces. The sticks were soaked in 2% lime water for 48–72 h until they were completely moistened, and then the excess moisture from the sticks was drained. The sticks were coated in a mixture of wheat bran (50%) and maize powder (50%). The sticks were then placed in bags parallel to each other, and the gaps among the sticks were filled with the BS mentioned in Section 2.3. The mixed substrates were compacted vertically in the bags. The filled bags were autoclaved at 123 °C for 2 h and then inoculated with mycelial plugs. The spawns were incubated at 25 °C in the dark. After each sample was completely covered with mycelia for 30–35 d, the samples were transferred to a mushroom chamber (39°9″ N, 116°3″ E) for 5–7 d after physiological maturation, and then grown adaptively for 2 d at a temperature of 20 ± 2 °C, relative humidity of 70–80%, and $CO_2$ concentration of 500–1000 ppm. Subsequently, the mushroom chamber was set to a temperature of 11–14 °C, a photoperiod of 12 h white light/dark (1500–2000 lux), approximately 90% relative humidity, and a $CO_2$ concentration of 1000–2000 ppm to induce differentiation of the primordium and development of the fruiting bodies. While the mushrooms were in their juvenile mushrooming stage, the growth conditions were adjusted to a temperature of 10–13 °C and $CO_2$ concentration at 7000–8000 ppm to promote the growth of stipes. The fruiting body agronomic traits,

including stipe length (cm), stipe diameter (cm), and pileus diameter (cm), were recorded using a slide caliper (Deli, Ningbo, Zhejiang, China). One flush of fruiting bodies was generated, harvested, and measured in the experiment, in line with the commercial cultural practices for *P. eryngii* in China. Fructification lasted 18–20 d. The fresh yields of fruiting bodies were weighed for 30 replicates. The biological efficiency (BE, %) was calculated by dividing the weight of the fresh yield of the fruiting bodies by the primary dry substrate weight in each bag [19].

*2.6. Composition Analysis*

After harvesting the fruiting bodies, the different samples were dried at 60 °C in an oven until the weight was constant, and then they were sealed for storage. The resulting powder was used for the determination of total proteins (%), fats (%), and ash (%), which were measured using standard procedures [20]. The sulfuric acid–anthrone colorimetric method was used to determine the total polysaccharide content of the mushrooms [21]. The crude protein content (N × 4.38) of the mushrooms was measured according to the macro Kjeldahl method [22]. The crude fat content of the mushrooms was determined by a Soxhlet extractor method (GB 5009.6-2016). The ash content of the mushrooms was analyzed by standard procedures (GB 5009.4-2016). The mineral elemental content of the fruiting bodies were determined by the following standard procedures: GB 5009.268-2016, GB/T 5009.92-2003, GB/T 5009.90-2003, GB/T 5009.14-2003, and GB/T 5009.91-2003. All test analyses were performed by the PONY Testing International Group (Beijing, China).

*2.7. Composition Statistics*

The original data were processed using Microsoft Excel (Microsoft Inc., Redmond, WA, USA). Differences among the means of groups were assessed using Duncan's multiple range tests at a 95% confidence level ($p < 0.05$). Statistical analyses were conducted using SPSS 19.0 (IBM Inc., Armonk, NY, USA).

## 3. Results

*3.1. Growth Rate of Mycelia*

The growth rate of mycelia is presented in Table 1. The fastest mycelial growth rate (2.99 mm d$^{-1}$) was recorded with the addition of 1% SSP. This condition was followed by a growth rate of 2.81 mm d$^{-1}$ in a 2% SSP growth medium. However, the growth medium without SSP resulted in a growth rate of only 2.64 mm d$^{-1}$. Higher concentrations of SSP caused no significant change in mycelial growth.

**Table 1.** Growth rate of *P. eryngii* cultivated on different substrates [#].

| Substrate | Growth Rates over Time (mm d$^{-1}$) | | | | Growth Rate (mm d$^{-1}$) |
|---|---|---|---|---|---|
| | 7~12 d | 12~17 d | 17~24 d | 24~31 d | |
| CK | 2.25 | 2.49 | 2.89 | 3.00 | 2.64 ± 0.14 d |
| CK1 | 2.72 | 2.55 | 2.72 | 2.82 | 2.70 ± 0.18 cd |
| T1 | 2.82 | 2.86 | 3.12 | 3.19 | 2.99 ± 0.14 a |
| T2 | 2.65 | 2.26 | 2.99 | 3.38 | 2.81 ± 0.18 bc |
| T3 | 2.53 | 2.35 | 2.82 | 2.77 | 2.62 ± 0.19 d |
| T4 | 2.47 | 2.43 | 3.06 | 3.70 | 2.79 ± 0.20 ab |
| T5 | 2.24 | 2.43 | 2.94 | 2.87 | 2.61 ± 0.10 d |

[#] Different lowercase letters denote significant differences in each column ($p < 0.05$). CK: treatment with sawdust 21%, sugarcane bagasse 21%, cottonseed hull 4.2%, ground corncobs 18.4%, wheat bran 18.4%, maize powder 6.8%, and soybean meal 8.4%; CK1: treatment with CK and 0.5% added CaCO$_3$; T1: treatment with CK and 1% added SSP; T2: treatment with CK and 2% added SSP; T3: treatment with CK and 3% added SSP; T4: treatment with CK and 4% added SSP; T5: treatment with CK and 5% added SSP.

*3.2. Characteristics of P. eryngii Fruiting Bodies on Different Substrates*

The highest average total fresh weight of the fruiting bodies (399.5 g bag$^{-1}$) was obtained in T2, followed by T1 (399.1 g bag$^{-1}$), T3 (393.8 g bag$^{-1}$), CK1 (390.1 g bag$^{-1}$),

T4 (361.1 g bag$^{-1}$), CK (349.8 g bag$^{-1}$), and T5 (306.0 g bag$^{-1}$). The biological efficiency of growth ranged from 64.8% to 84.6%, with the highest value recorded after supplementation with 2% SSP substrate (Table 2). Fruiting body weight and biological efficiency were significantly different ($p < 0.05$) among the samples evaluated, but there was no significant difference in yields in CK1, T1, T2, and T3. The yields of the above four treatments were higher than those achieved using substrates not supplemented with SSP. The length of the fruiting body, thickness of the stipe, and the diameter of the pileus of CK1, T1, T2, T3, and T4 were all longer than those of CK.

**Table 2.** Agronomic traits of *P. eryngii* cultivated on different substrates [#].

| Substrate | Fresh Fruiting Body Yield (g bag$^{-1}$) | Biological Efficiency (%) | Length of Fruit Body (cm) | Thickness of Stipe (cm) | Diameter of Pileus (cm) |
|---|---|---|---|---|---|
| CK | 349.8 ± 38.3 b | 74.0 ± 7.8 b | 14.9 ± 1.4 b | 5.4 ± 0.6 a | 4.5 ± 0.6 de |
| CK1 | 390.1 ± 40.5 a | 82.8 ± 8.6 a | 16.4 ± 1.5 a | 5.5 ± 0.6 a | 4.7 ± 0.6 cd |
| T1 | 399.1 ± 41.2 a | 84.5 ± 8.7 a | 16.0 ± 1.5 a | 5.7 ± 0.8 a | 5.4 ± 0.7 a |
| T2 | 399.5 ± 44.2 a | 84.6 ± 9.4 a | 16.2 ± 1.6 a | 5.4 ± 0.9 a | 5.2 ± 0.7 ab |
| T3 | 393.8 ± 35.2 a | 83.3 ± 7.4 a | 16.3 ± 2.0 a | 5.7 ± 0.5 a | 4.9 ± 0.8 bc |
| T4 | 361.1 ± 34.1 b | 76.4 ± 7.2 b | 15.7 ± 1.6 a | 5.6 ± 0.6 a | 4.8 ± 0.5 cd |
| T5 | 306.0 ± 45.0 c | 64.8 ± 9.5 c | 13.6 ± 1.5 c | 5.5 ± 0.6 a | 4.3 ± 0.5 e |

[#] Different lowercase letters denote significant differences in each column ($p < 0.05$). CK: treatment with sawdust 21%, sugarcane bagasse 21%, cottonseed hull 4.2%, ground corncobs 18.4%, wheat bran 18.4%, maize powder 6.8%, and soybean meal 8.4%; CK1: treatment with CK and 0.5% added CaCO$_3$; T1: treatment with CK and 1% added SSP; T2: treatment with CK and 2% added SSP; T3: treatment with CK and 3% added SSP; T4: treatment with CK and 4% added SSP; T5: treatment with CK and 5% added SSP.

### 3.3. Nutrient Content of Mushrooms

The variations in the major nutrient content of the king oyster mushrooms were observed to be related to the composition of the cultivation substrates (Table 3). The crude fiber content of the dried king oyster mushrooms were significantly higher (7.81 g 100 g$^{-1}$) when grown on T3 than on all other substrates, and the lowest nutrient content (5.21 g 100 g$^{-1}$) of crude fiber was obtained in the controls (i.e., CK and CK1). The highest nutrient value (2.11 g 100 g$^{-1}$) for fat content was obtained in the T1 treatment, which showed a value that was significantly different from those of mushrooms grown on all other substrates. Compared with the test samples, the lowest value (1.32 g 100 g$^{-1}$) of fat content was obtained with T3. The range of ash content was from 6.87% to 8.57% and was significantly different for all substrates. The highest value of crude protein (23.85 g 100 g$^{-1}$) was obtained in T4, and the lowest value (21.15 g 100 g$^{-1}$) was obtained in T1 ($p < 0.05$). The crude polysaccharide content was found to be the highest (4.96 g 100 g$^{-1}$) on the T2 substrate, and the minimum value (4.18 g 100 g$^{-1}$) was recorded from T3 substrate.

**Table 3.** Nutritional value of *P. eryngii* fruiting bodies cultivated on different substrates [#]* (g 100 g$^{-1}$).

| Substrate | Fiber | Fat | Ash | Protein | Polysaccharide |
|---|---|---|---|---|---|
| CK | 5.21 ± 0.02 g | 1.71 ± 0.01 c | 5.13 ± 0.02 c | 21.55 ± 0.15 d | 4.26 ± 0.01 d |
| CK1 | 7.40 ± 0.01 b | 1.66 ± 0.01 d | 5.20 ± 0.01 b | 22.65 ± 0.25 c | 3.43 ± 0.02 f |
| T1 | 6.34 ± 0.04 d | 2.11 ± 0.01 a | 4.88 ± 0.02 e | 21.15 ± 0.25 d | 4.54 ± 0.08 b |
| T2 | 5.75 ± 0.04 f | 1.45 ± 0.01 f | 5.05 ± 0.02 d | 22.85 ± 0.25 bc | 4.96 ± 0.04 a |
| T3 | 7.81 ± 0.04 a | 1.32 ± 0.01 g | 5.03 ± 0.07 d | 22.55 ± 0.05 c | 4.18 ± 0.01 e |
| T4 | 6.54 ± 0.01 c | 1.90 ± 0.01 b | 5.32 ± 0.02 a | 23.85 ± 0.35 a | 4.43 ± 0.02 c |
| T5 | 5.90 ± 0.03 e | 1.62 ± 0.01 e | 5.28 ± 0.02 a | 23.25 ± 0.25 b | 4.28 ± 0.03 d |

[#] Different lowercase letters denote significant differences in each column ($p < 0.05$). * Dry matter. CK: treatment with sawdust 21%, sugarcane bagasse 21%, cottonseed hull 4.2%, ground corncobs 18.4%, wheat bran 18.4%, maize powder 6.8%, and soybean meal 8.4%; CK1: treatment with CK and 0.5% added CaCO$_3$; T1: treatment with CK and 1% added SSP; T2: treatment with CK and 2% added SSP; T3: treatment with CK and 3% added SSP; T4: treatment with CK and 4% added SSP; T5: treatment with CK and 5% added SSP.

### 3.4. Macronutrient Elements of Mushrooms

The mineral content of the mushrooms grown on all substrates supplemented with SSP had favorable concentrations of P, K, Mg, Na, and Ca (Table 4). All fruiting bodies grown on substrates supplemented with SSP had higher amounts of both P and Na than those grown on CK and CK1. The fruiting bodies grown on T2 had higher Ca content and lower K and Mg content than those grown on CK and CK1. Mushrooms cultivated on the sawdust and sugarcane bagasse medium without additional Ca supplementation showed $41.0 \pm 0.3$ mg kg$^{-1}$ of Ca in the dried product fruiting bodies. However, the Ca content in the fruiting body increased to a level of $67.2 \pm 0.8$ mg kg$^{-1}$ by supplementation with 2% SSP. The substrates supplemented 0.5% CaCO$_3$ generated the P, K, or Mg content in fruiting bodies at a same level as that grown on 2% SSP substrate, but the Na content were the lowest in the former formula.

**Table 4.** The macronutrient elements in fruiting bodies cultivated on different substrates [#]*.

| Treatments | CK | CK1 | T1 | T2 | T3 | T4 | T5 |
|---|---|---|---|---|---|---|---|
| Phosphorus (mg kg$^{-1}$) | $736.0 \pm 4.2$ d | $741 \pm 5.0$ d | $766.0 \pm 8.5$ c | $738.0 \pm 22.6$ d | $744.5 \pm 20.5$ bc | $791.5 \pm 0.7$ ab | $797.5 \pm 24.7$ a |
| Potassium (g kg$^{-1}$) | $25.4 \pm 0.02$ bc | $25.9 \pm 0.00$ b | $25.7 \pm 0.06$ b | $24.5 \pm 0.06$ bc | $23.2 \pm 0.00$ d | $27.4 \pm 0.18$ a | $25.6 \pm 0.04$ b |
| Magnesium (g kg$^{-1}$) | $1.10 \pm 0.03$ c | $1.09 \pm 0.03$ c | $1.17 \pm 0.00$ b | $1.09 \pm 0.01$ c | $1.21 \pm 0.01$ b | $1.30 \pm 0.08$ a | $1.30 \pm 0.01$ a |
| Sodium(mg kg$^{-1}$) | $205.0 \pm 1.4$ d | $190 \pm 0.0$ e | $230.0 \pm 0.0$ c | $229.5 \pm 14.8$ c | $228.0 \pm 4.2$ c | $298.0 \pm 21.2$ a | $250.5 \pm 7.8$ b |
| Calcium (mg kg$^{-1}$) | $41.0 \pm 0.3$ e | $63.7 \pm 0.6$ b | $41.0 \pm 0.4$ e | $67.2 \pm 0.8$ a | $48.6 \pm 0.4$ d | $50.2 \pm 0.1$ d | $52.3 \pm 2.8$ c |

[#] Different lowercase letters mean significant differences in each row ($p < 0.05$). * Dry matter. CK: treatment with sawdust 21%, sugarcane bagasse 21%, cottonseed hull 4.2%, ground corncobs 18.4%, wheat bran 18.4%, maize powder 6.8%, and soybean meal 8.4%; CK1: treatment with CK and 0.5% added CaCO$_3$; T1: treatment with CK and 1% added SSP; T2: treatment with CK and 2% added SSP; T3: treatment with CK and 3% added SSP; T4: treatment with CK and 4% added SSP; T5: treatment with CK and 5% added SSP.

### 3.5. Micronutrient Elements of Mushrooms

The micronutrient content of fruiting bodies grown on all substrates have been tested (Table 5). The amount of Mn was lower in all fruiting bodies grown with calcium carbonate and SSP, ranging from 5.58 to 6.27 mg kg$^{-1}$, compared with those on the control substrate (6.77 mg kg$^{-1}$). The iron (Fe) content in all fruiting bodies grown with SSP (ranging from 34.6 to 44.1 mg kg$^{-1}$) was higher than those grown on CK (32.7 mg kg$^{-1}$) and CK1 (29.3 mg kg$^{-1}$). The zinc (Zn) content was highest (91.3 mg kg$^{-1}$) in the samples grown with 5% SSP, with a 13.8% increase relative to the control substrate (80.2 mg kg$^{-1}$). The Zn content varied across the various growth substrates, with no specific trend. The aluminum (Al) content was highest (5.84 mg kg$^{-1}$) in the samples grown with 4% SSP, which showed an increase of 28.9% compared with the control substrate (4.53 mg kg$^{-1}$). The boron (B) content was highest (21.6 mg kg$^{-1}$) in the samples grown with 4% SSP, which was significantly higher than that in the control substrate (19.6 mg kg$^{-1}$).

**Table 5.** The micronutrient elements in fruiting bodies cultivated on different substrates [#]*.

| Treatments | CK | CK1 | T1 | T2 | T3 | T4 | T5 |
|---|---|---|---|---|---|---|---|
| Manganese (mg kg$^{-1}$) | $6.77 \pm 0.04$ a | $6.35 \pm 0.20$ b | $5.58 \pm 0.04$ de | $5.99 \pm 0.42$ bc | $5.44 \pm 0.11$ e | $5.87 \pm 0.13$ cd | $6.27 \pm 0.37$ b |
| Iron (mg kg$^{-1}$) | $32.7 \pm 0.5$ e | $29.3 \pm 0.6$ f | $34.6 \pm 0.0$ cd | $35.1 \pm 0.5$ d | $36.6 \pm 1.8$ c | $44.1 \pm 0.2$ a | $39.2 \pm 1.5$ b |
| Zinc (mg kg$^{-1}$) | $80.2 \pm 1.3$ c | $72.7 \pm 3.2$ e | $82.0 \pm 1.8$ c | $76.4 \pm 2.5$ d | $87.8 \pm 3.0$ b | $80.5 \pm 3.5$ c | $91.3 \pm 0.4$ a |
| Boron (mg kg$^{-1}$) | $19.6 \pm 0.5$ b | $15.1 \pm 0.0$ d | $18.8 \pm 0.8$ bc | $14.2 \pm 0.4$ e | $18.6 \pm 0.7$ c | $21.6 \pm 1.1$ a | $21.4 \pm 0.6$ a |
| Aluminum (mg kg$^{-1}$) | $4.53 \pm 0.25$ d | $4.09 \pm 0.00$ e | $5.00 \pm 0.23$ b | $4.73 \pm 0.16$ c | $4.12 \pm 0.08$ e | $5.84 \pm 0.05$ a | $5.07 \pm 0.12$ b |

[#] Different lowercase letters mean significant differences in each row ($p < 0.05$). * Dry matter. CK: treatment with sawdust 21%, sugarcane bagasse 21%, cottonseed hull 4.2%, ground corncobs 18.4%, wheat bran 18.4%, maize powder 6.8%, and soybean meal 8.4%; CK1: treatment with CK and 0.5% added CaCO$_3$; T1: treatment with CK and 1% added SSP; T2: treatment with CK and 2% added SSP; T3: treatment with CK and 3% added SSP; T4: treatment with CK and 4% added SSP; T5: treatment with CK and 5% added SSP.

## 4. Discussion

### 4.1. Influence of SSP Supplementation on the Growth Rate of Mycelia

According to Table 1, *P. eryngii* had the fastest mycelial growth rate when it was grown on T1, which was supplemented with 1% SSP. However, with greater additions of SSP, the growth rate of mycelium slowed down. When the content of SSP was 5%, the growth rate of mycelium was lower than CK and CK1. These findings are in good agreement with a study of *P. florida* PF05 grown on wheat straw supplemented with oyster shell powder. In the previous work, the mycelial growth rate was related to shell powder supplementation. Minerals, including magnesium (Mg), Ca, Fe, copper (Cu), manganese (Mn), Zn, and often molybdenum (Mo), are required by edible fungi for growth. Variations in the substrate and intracellular Ca concentrations can cause a variety of different reactions on growth, differentiation, and sporulation [22]. In general, shell powder predominantly contains $CaCO_3$; however, there are often other minerals such as, P, Mn, Zn, and Fe [3]. Therefore, these low levels of micronutrients from SSP may also contribute to the growth of mycelia. However, prior work has also found that higher levels of supplementation with shell powders can suppress or kill mycelia [23,24]. A previous report showed that the mycelial growth of *Hypsizigus marmoreus* slightly increased with the addition of $CaCO_3$ to a sawdust medium; however, mycelial growth was completely inhibited on potato sucrose agar medium supplemented with $CaCO_3$ [25]. Thus, it can be speculated that supplementation with a low concentration of SSP aids mycelial growth by providing an important source of $CaCO_3$.

### 4.2. Influence of SSP Supplementation on the Morphological and Fruit Characteristics of P. eryngii

This study demonstrated that substrates containing up to 3% SSP or 0.5% $CaCO_3$ were more productive than substrates of the control. The yield of the fruiting bodies and the biological efficiency of *P. eryngii* increased following the addition of shell powder, but beyond a certain concentration, they declined. When the concentration of shell powder was 5%, the yield of the fresh fruiting bodies was lower than those with no SSP supplementation. Our findings generally agreed with a prior report by Naraian, which found that the highest biological efficiency was achieved upon supplementation with 3% oyster shell powder [14]. A BS with 1% lime and 1% gypsum is the most popular basal substrate for producing *P. eryngii* in China, with a BE of 75.6% [26]. These data are consistent with the hypothesis that the composition of the growth substrate has a strong influence on mushroom yield, and $CaCO_3$ is necessary for improving the yield.

The morphological characteristics of *P. eryngii* fruiting bodies grown on the different substrates were remarkable different (Figure 1 and Table 2). The range of the pileus diameter was 4.3–5.4 cm. The lengths of fruiting bodies harvested from the treatment groups with SSP were longer than those from CK but showed no difference from those from CK1. However, the thickness of the stipes collected from the T1 to T5 groups with SSP was not significantly different from that of the control group. The quality of edible mushroom is determined by a combination of shape, weight, and color; therefore, these are likely to be good criteria for evaluating mushrooms. The results mentioned above indicate that based on the yield of the fruiting bodies, biological efficiency, pileus diameter, stipe diameter, and length, the substrate with 2% SSP supplementation has the potential to be an excellent medium for the growth of *P. eryngii*.

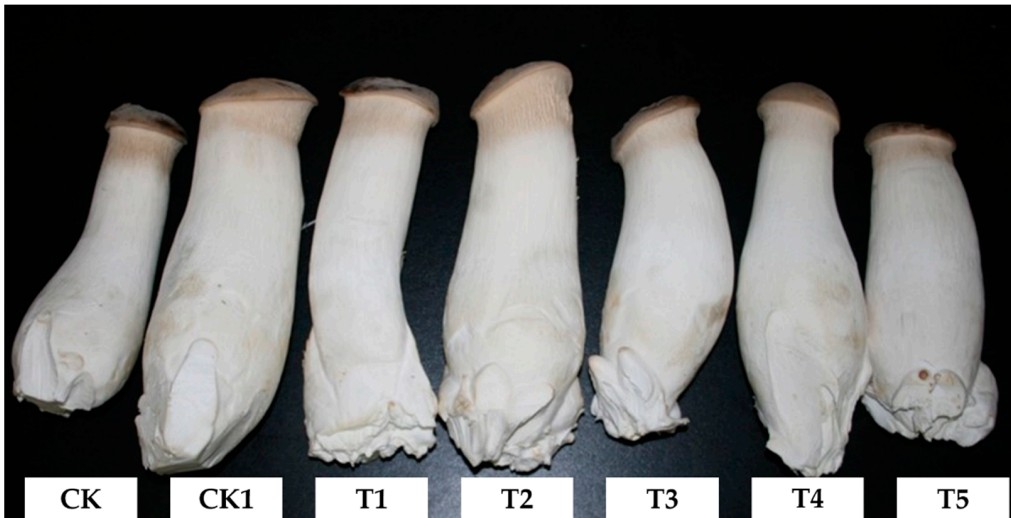

**Figure 1.** Morphological characteristics of *P. eryngii* fruiting bodies from mycelia grown on the different substrates. CK: treatment with sawdust 21%, sugarcane bagasse 21%, cottonseed hull 4.2%, ground corncobs 18.4%, wheat bran 18.4%, maize powder 6.8%, and soybean meal 8.4%; CK1: treatment with CK and 0.5% added $CaCO_3$; T1: treatment with CK and 1% added SSP; T2: treatment with CK and 2% added SSP; T3: treatment with CK and 3% added SSP; T4: treatment with CK and 4% added SSP; T5: treatment with CK and 5% added SSP.

*4.3. Influence of SSP Supplementation on the Nutrient Content of Mushrooms*

According to previous work [27], the content of polysaccharides in mushroom is closely related to the composition of the growth substrate. Supplementation with SSP appeared to improve polysaccharide accumulation in *P. eryngii* compared with mushrooms grown on a control substrate; however, high concentrations of SSP inhibited polysaccharide accumulation. Previous studies of mushroom chemical content by Ryu and Oyetayo demonstrated that the major nutrient content of edible mushrooms can be influenced by the substrates on which they are cultivated [28,29]. This means that the availability of nutrients may affect the composition and, particularly, the protein content of the fruiting bodies of oyster mushrooms. According to Akiyama, scallop shell protein analysis has revealed the presence of several amino acids, mainly valine, alanine, glutamine, serine, proline, leucine, tyrosine, lysine, etc. [30]; therefore, these amino acids present in scallop shells may provide nutrients to fungi and help in their protein synthesis. At the same time, the other ingredients of fruiting bodies may also change in addition to the proteins. Our results are consistent with the above-mentioned observation despite the different mushroom species tested. In addition to high yield, BE, and excellent agronomic traits, the higher crude protein content and lower fat content found in king oyster mushrooms grown with 2% SSP supplementation support SSP addition as a new strategy to produce superior quality *P. eryngii* to satisfy the demands of consumers.

*4.4. Influence of SSP Supplementation on the Macronutrient Elements of Mushrooms*

Large variations exist in the maneral elements of mushrooms depending on their genetic variations, cultivation technologies, and growth substrate composition [31,32]. The Ca content in the fruiting bodies of *P. eryngii* tended to improve with increasing Ca content in the substrate as was expected. However, the substrate that contained more than 5% of SSP significantly decreased the yield of the fruiting bodies, suggesting a maximum limit to supplementation-based growth enhancement (Table 2). Our findings are similar with the results of the effect of adding calcined scallop shells to sawdust medium on the cultivation of *P. eryngii* [12,13]. This might be due to the fact that high levels of supplementation with SSP can suppress or kill mycelia, leading to lower yields.

The Ca content of *P. eryngii* increased by 64% with the addition of 2% SSP as a Ca source compared with the control, and this Ca content was higher than that of the CK1 treatment, which added 0.5% $CaCO_3$. Meanwhile, CK1 was also found to have high Ca concentrations in mushroom but lower quantities in substrate than with SSP. It might be possible that the mycelial absorption of $CaCO_3$ from SSP is lower than supplemented pure $CaCO_3$ [2,3]. The T2 treatment also produced a greater yield of mushroom (Table 2). Our cumulative findings suggest that 2% SSP supplementation produces *P. eryngii* containing higher Ca content. Before industrial application, however, further studies are needed to examine the effects of smaller changes in SSP content (e.g., 1.8% versus 1.9% versus 2.0%).

*4.5. Influence of SSP Supplementation on the Micronutrient Elements of Mushrooms*

The content of micronutrient elements are shown in Table 5. The macronutrient elements of mushrooms, including Mn, Cu, and Zn, are cofactors of numerous enzymes, and their concentrations in healthy tissues are highly regulated [32,33]. However, these standard values are lower than those recommended by China's food safety standard, GB 2762-2012. The content of Mn in most mushrooms is approximately one-tenth that of Ca [14]. Zn is an essential trace element for human nutrition that is an integral part of many enzyme systems including the DNA polymerase complex. Zn deficiency has been associated with the inhibition of growth and sexual immaturity [34]. In a prior study of wild Chinese mushroom samples, large variations in Zn values were noted, ranging from 20 to 140 mg kg$^{-1}$ [35].

Similar studies have shown that the content of Al and B (mg kg$^{-1}$, dry weight) in 34 wild mushrooms ranged from 53.96 to 3308 and 0.229 to 46.93, respectively. The Al content was the highest in *Pleurotus ostreatus* (3308 mg kg$^{-1}$) and the lowest in *Laetiporus sulphureus* (53.96 mg kg$^{-1}$) [36]. It has been shown that wild mushrooms can contain high concentrations of metals, but the macronutrient elemental content in cultivated mushrooms are typically low [37]. In this work, there was no significant logical trend in the relative quantities of the different mineral nutrients with increasing SSP, which means that there is no simple correlation between them. So it can not be predicted to have any effect on those parameters.

**5. Conclusions**

SSP can efficiently supply nutrients to improve the development and yield of mushrooms. The utilization of SSP for cultivation of *P. eryngii* is promising and has potential commercial applications in the mushroom industry. Our results show that substrates supplemented with 2% SSP resulted in the highest mushroom fresh weight and biological efficiency, and the fruiting bodies showed excellent agronomic traits by exhibiting improved protein, fiber, polysaccharide, micro- and macronutrient content. Further studies are needed to identify the effects of other forms and concentrations of Ca on the yield, biological efficiency, and nutritional content of edible fungi. Meanwhile, substrates supplemented with $CaCO_3$ were used as controls, and attention should be given to impurities. From the above results, it can be concluded that $CaCO_3$ improved the yield or BE at the same level as that of 2% SSP. The only significant difference was in the diameters of the caps. In addition, it may be essential to measure the amount of Ca (or other minerals including Se and Zn) present in the substrates, which can affect the mineral concentrations of the harvested mushrooms depending on the various Ca sources and concentrations. Increased mineral concentrations can enhance the nutritional value of mushrooms and, therefore, may help increase consumer demand. SSP may, thus, be a useful means for substrate supplementation that can allow for the sustainable commercial cultivation of *P. eryngii* over the long term. Therefore, SSP can be used as an alternative to other Ca sources in cultivation substrates.

**Author Contributions:** Conceptualization, Y.Z. (Yajie Zou) and Q.H.; methodology, Y.Z. (Yajie Zou); software, H.Z.; validation, Y.Z. (Yuanyuan Zhou) and Z.L.; data curation, H.Z.; writing—original draft preparation, Y.Z. (Yuanyuan Zhou); writing—review and editing, Y.Z. (Yajie Zou); visualization, Y.Z. (Yuanyuan Zhou); supervision and project administration, Y.Z. (Yajie Zou); project administration and funding acquisition, Y.Z. (Yajie Zou) and Q.H. All authors have read and agreed to the published version of the manuscript.

**Funding:** This research was funded by the Fundamental Research Funds for Central Non-profit Scientific Institution (grant number: 2014-21), the Special Fund for Agro-Scientific Research in the Public Interest, China (grant number: 201503137), and the China Agriculture Research System (grant number: CARS20).

**Institutional Review Board Statement:** Not applicable.

**Informed Consent Statement:** Not applicable.

**Data Availability Statement:** Not applicable.

**Conflicts of Interest:** The authors declare no conflict of interest.

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
