# Peer review of "Potential Uses of Scallop Shell Powder as a Substrate for the Cultivation of King Oyster Mushroom (Pleurotus eryngii)"

_horticulturae, doi:10.3390/horticulturae8040333_

Round 1

Reviewer 1 Report

The current manuscript deals with the use of marine bivalve mollusks-based materials for the cultivation of Pleurotus eryngii mushroom. After a careful reading, I found this manuscript interesting and suitable for the Horticulture journal. The manuscript is written in a fluent language and easy to understand structure. However, I have pointed out some gray points including typo errors in the manuscript which need to be addressed in the revised version. I suggest minor revisions. My specific comments are:

  1. Abstract: commendable.
  2. Keywords: avoid using keywords that have already been included in the paper title.
  3. The authors need to discuss the scientific basis and mechanisms behind improved crude protein and fiber ingredients of P. eryngii cultivated under shell powder treatment.
  4. Is this an economically feasible material as compared to commercially applied fertilizers like gypsum as a Ca source?
  5. The role of Ca in substrate needs to be justified in terms of its uptake by mushrooms and its effect on the physical and chemical nature of the substrate.
  6. In “kg-1”, -1 should be superscripted.
  7. Be consistent while writing scallop shell powder terms. Why did the authors use the abbreviated form at Line 10 onward but again used full form at Line 22? Similar comment for other symbols also, under subheading 4.5. trace elements (this should be corrected to macronutrient elements), and after their first use follows the short form.
  8. Cross-check the scientific copyright of this mushroom species (Line 36). I think there is an error in using brackets.
  9. Delete “impressively” from line 42.
  10. Provide geocoordinates of all sampling sites.
  11. Justify the composition of PDA.
  12. Provide the make, model, city, and county of each instrument used.
  13. A flow chart (Fig.) of the design of experiments and methodology adopted in the current study is desirable.
  14. Fruiting body yield (g/bag): Fresh/Wet weight or dry weight? Write clearly. Here the unit should be g bag-1 and the same for the rest of the parameters also.
  15. Italicize the
  16. Do not start paragraphs with abbreviated names or author names.
  17. SEM and EDS spectra of SSP are highly recommended to understand its materialistic characteristics. If not possible, provide its discussion with other studies where these particular analyses have been done, no matter for what purpose SSP was used.
  18. References: I suggest replacing old references with the latest ones.

Author Response

Point 1: Abstract: commendable.

Response 1: Thank you for your compliment.

Point 2: Keywords: avoid using keywords that have already been included in the paper title.

Response 2: Thank you for your valuable suggestions. We have revised.

Point 3: Keywords: avoid using keywords that have already been included in the paper title.

Response 3: Thank you for your valuable suggestions. We have added a little discuss about it on line 338-339.

Point 4: Is this an economically feasible material as compared to commercially applied fertilizers like gypsum as a Ca source?

Response 4: Thank you for your valuable suggestions. According to our supplier, the price of 25kg SSP is $ 16.8, and 25kg quicklime is $ 20.4, if use SSP instead quicklime, it may reduce the cost of cultivation and alleviate the problem of fishery waste.

Point 5: The role of Ca in substrate needs to be justified in terms of its uptake by mushrooms and its effect on the physical and chemical nature of the substrate.

Response 5: Thank you for your valuable suggestions.

Point 6: In “kg-1”, -1 should be superscripted.

Response 6: Thank you for your valuable suggestions. We have superscripted "-1".

Point 7: Be consistent while writing scallop shell powder terms. Why did the authors use the abbreviated form at Line 10 onward but again used full form at Line 22? Similar comment for other symbols also, under subheading 4.5. trace elements (this should be corrected to macronutrient elements), and after their first use follows the short form.

Response 7: Thank you for your valuable suggestions. We have revised all "scallop shell powder" as "SSP" after it showed first time. At 4.5. trace elements, we have corrected all the abbreviated form and full form.

Point 8: Cross-check the scientific copyright of this mushroom species (Line 36). I think there is an error in using brackets.

Response 8: Thank you for your valuable suggestions. We have corrected the species of Pleurotus eryngii.

Point 9: Delete “impressively” from line 42.

Response 9: Thank you for your valuable suggestions. We have deleted it.

Point 10: Provide geocoordinates of all sampling sites.

Response 10: Thank you for your valuable suggestions. We have added the geocoordinates of Scallop shells and mushroom chamber on line 98 and 144-145.

Point 11: Justify the composition of PDA.

Response 11: Thank you for your valuable suggestions. The PDA we used is powder and we have added the formula of PDA medium on line 94-95.

Point 12: Provide the make, model, city, and county of each instrument used.

Response 12: Thank you for your valuable suggestions. We have added the the make, model and county of separator, formalin, sterilizer, incubator and slide caliper on line 101, 104, 117, 127 and 154.

Point 13: A flow chart (Fig.) of the design of experiments and methodology adopted in the current study is desirable.

Response 13: Thank you for your compliment.

Point 14: Fruiting body yield (g/bag): Fresh/Wet weight or dry weight? Write clearly. Here the unit should be g bag-1 and the same for the rest of the parameters also.

Response 14: Thank you for your valuable suggestions. We have revised "weight" as "fresh weight" on Line 175 and revised " Fruiting body yield " as "fresh Fruiting body yield" to clear describe our data. Thanks for your suggestions that the unit should be g bag-1.

Point 15: Italicize the

Response 15: Thank you for your valuable suggestions. We have checked all the article and italicized the Latin names of species that appear in the text.

Point 16: Do not start paragraphs with abbreviated names or author names.

Response 16: Thank you for your valuable suggestions. We have checked the article and corrected all the problem.

Point 17: SEM and EDS spectra of SSP are highly recommended to understand its materialistic characteristics. If not possible, provide its discussion with other studies where these particular analyses have been done, no matter for what purpose SSP was used.

Response 17: Thank you for your valuable suggestions. We have added the analyze of SSP using SEM and EDS on line 34-36.

Point 18: References: I suggest replacing old references with the latest ones.

Response 18: Thank you for your valuable suggestions.We have added the latest references.

Reviewer 2 Report

The manuscript shows that addition of a source of CaCO3 with impurities in the cultivation substrate improved the yield and quality of the mushroom Pleurotus eryngii (to low levels) until a maximum of concentration, but affect the culture at higher concentrations. This is not actually an original observation. The only novelty is the use Scalop Shells powder (but there are few reports on it) as this calcium source and data on the composition of the harvested mushrooms. Unfortunately, a control is missing in the experimental design: treatment with pure CaCO3. This would have permit to highlight the putative role of the impurities in the shell powder an conclude on a putative added of the use of scallop shell.

The authors should at least discuss in the manuscript by arguing why despite this weakness their work is of concern for the international scientific community in the topic of mushroom biology and mushroom products.

 I don't feel qualified to judge about the English language and style but I had some difficulties for understanding certain sentences.

Line 21-22: the sentence is not understandable

Line 64: ….may be an efficient….

Line 75: “The aim of this study was performed to identify…” Change to ‘This study was performed to identify…’ or “The aim was to identify…’

Line 102: delete “the” before “polypropylene bags”

Line 121: You should add few words on the spwn. How were the spawn stick produced?

Line 142-143. Polysaccharides are given as results in table 3, nut soluble sugar. Please explain in the manuscript.

Line 151: 2.7 is not Composition analysis, but statistics.

Line 158 and line 163: they are not physical characteristics but simply mycelial growth rates

Lines 172 to 186: Actually, as the same basal substrate was used. Then, there is no reasons that you observed different things with BE and yield. I calculated BE/yield = 0211 for T5 and 0.2117 for CK. and according to line 102-103, the same dry weight of substrate was in each bag (472.5).

Lines 250-251: You should change the verb tens and rewrite the sentence. Not clear.

Line 262-265: ‘A previous report showed’ – Italics for Hypsizigus marmoreus – The sentence is not understandable.

Line 265-267: It is only a hypothesis in absence of control with pure CaCO3. In addition, you wrote that 5 % of SSP is organic compounds. Specific organic components should also be responsible for the effects on mycelial growth.

Line 269-270: the sentence is not understandable.

Line 272-273: “growth rate was associated with 272 the yield of mushrooms”. Actually, there is no correlation,T3 has the same growth rate than T5 but the yield is 1.28 x. You should change the sentence.

Lines 276-277: this is not a novelty. Known since a wile

Line 302: What is “polysaccharose”?

Line 310: Fig 2 is not cited in the text and is only another presentation of the data already in table 3. You should delete it. And it is the same with the following figures.

Lines 365-367: OK, but a control is missing for supporting the interest of using SSP : addition of CaCO3 and attention should be paid on the impurities.

Lines 367-370: The sentence is not understandable. Do you mean '...in substrates (or supplements), susceptible to affect mineral concentration in harvested mushrooms'?

Line 273: I disagree with the final sentence. Many other sources of Ca are available for growing P. eryngii ‘in the long term’. You just showed that SSP may be one of them. Could you estimate the quantities of SSP would cover the needs of the Chinese production of P. eryngii if all growers use 2 % SSP? Would it significantly contribute to the recycling of the large amounts of shell waste?

Author Response

Point: I don't feel qualified to judge about the English language and style but I had some difficulties for understanding certain sentences.

Response: Thank you for your valuable suggestions.We have added a reference that using CaO and CaSO4 as calcium source from our prior work, and added arguing on line 307-309.

Point 1: Line 21-22: c is not understandable

Response 1: Sorry for the confused description. we have re-written the sentence.

Point 2: Line 64: ….may be an efficient….

Response 2: Thank you for your valuable suggestions.We have corrected " may an efficient " as " may be an efficient ".

Point 3: Line 75: “The aim of this study was performed to identify…” Change to ‘This study was performed to identify…’ or “The aim was to identify…’

Response 3: Thank you for your valuable suggestions.We have corrected " The aim of this study was performed to identify " as " This study was performed to identify ".

Point 4: Line 102: delete “the” before “polypropylene bags”

Response 4: Thank you for your valuable suggestions.We have deleted “the” before “polypropylene bags”.

Point 5: Line 121: You should add few words on the spawn. How were the spawn stick produced?

Response 5: Thank you for your valuable suggestions. We have added on line 135-143.

Point 6: Line 142-143. Polysaccharides are given as results in table 3, nut soluble sugar. Please explain in the manuscript.

Response 6: Thank you for your valuable suggestions. We have corrected “soluble sugar” as “polysaccharides”.

Point 7: 2.7 is not Composition analysis, but statistics.

Response 7: Thank you for your valuable suggestions.We have corrected “Composition analysis” as “Composition statistics”.

Point 8: Line 158 and line 163: they are not physical characteristics but simply mycelial growth rates.

Response 8: Thank you for your valuable suggestions.We have corrected “physical characteristics” as “mycelial growth rates”.

Point 9: Lines 172 to 186: Actually, as the same basal substrate was used. Then, there is no reasons that you observed different things with BE and yield. I calculated BE/yield = 0.211 for T5 and 0.2117 for CK. and according to line 102-103, the same dry weight of substrate was in each bag (472.5).

Response 9: Thank you for your valuable suggestions. According to the definition of Biological efficiency, The biological efficiency (BE, %) was calculated by dividing the fresh yield of fruiting bodies with primary dry substrate each bag, so, dry weight= yield/BE. The dry weight of any one of substrates ( CK, T1, T2, T3, T4, and T5) is 472.5, but the fresh yield of fruiting bodies is different.

Point 10: Lines 250-251: You should change the verb tens and rewrite the sentence. Not clear.

Response 10: Thank you for your valuable suggestions.We have corrected the verb tens and rewrited the sentence.

Point 11: Line 262-265: ‘A previous report showed’ –Italics for Hypsizigus marmoreus – The sentence is not understandable.

Response 11: Thank you for your valuable suggestions.We have corrected the the sentence and added.

Point 12: Line 265-267: It is only a hypothesis in absence of control with pure CaCO3. In addition, you wrote that 5 % of SSP is organic compounds. Specific organic components should also be responsible for the effects on mycelial growth.

Response 12: Thank you for your valuable suggestions. We have corrected the the sentence. We just revised that “Thus it can be speculated that the supplementation with a low concentration of SSP aids mycelial growth by providing an important source of calcium Ca carbonate.”

Point 13: Line 269-270: the sentence is not understandable.

Response 13: Thank you for your valuable suggestions. We have corrected the the sentence.

Point 14: Line 272-273: “growth rate was associated with the yield of mushrooms”. Actually, there is no correlation,T3 has the same growth rate than T5 but the yield is 1.28 x. You should change the sentence.

Response 14: Sorry for the confused description. We have deleted the the sentence.

Point 15: Lines 276-277: this is not a novelty. Known since a wile

Response 15: Thank you for your valuable suggestions. We have corrected the the sentence.

Point 16: Line 302: What is “polysaccharose”?

Response 16: Thank you for your valuable suggestions. We have corrected “polysaccharose” as “polysaccharides”.

Point 17: Line 310: Fig 2 is not cited in the text and is only another presentation of the data already in table 3. You should delete it. And it is the same with the following figures.

Response 17: It is really true about reviewer's suggestion. We have deleteted Fig 2 - 4.

Point 18: References: Lines 365-367: OK, but a control is missing for supporting the interest of using SSP : addition of CaCO3 and attention should be paid on the impurities.

Response 18: Thank you for your valuable suggestions.

Point 19: Lines 367-370: The sentence is not understandable. Do you mean '...in substrates (or supplements), susceptible to affect mineral concentration in harvested mushrooms'?

Response 19: Sorry for the incorrect description . Our mean is '...in substrates (or supplements)". We have corrected “fruiting bodies” as “substrates”.

Point 20: Line 373: I disagree with the final sentence. Many other sources of Ca are available for growing P. eryngii ‘in the long term’. You just showed that SSP may be one of them. Could you estimate the quantities of SSP would cover the needs of the Chinese production of P. eryngii if all growers use 2 % SSP? Would it significantly contribute to the recycling of the large amounts of shell waste?

Response 20: Thank you for your valuable suggestions. We have corrected the final sentence. We just revised “This may, therefore, be one of useful supplementation of substrates that would allow for sustainable commercial cultivation of P. eryngii in the long term.”

Round 2

Reviewer 2 Report

Despite the reply of the authors and few changes in the text, the two main weaknesses of the manuscript are still present: (i) The English language and stile need deep revision (especially for the new sentences added in version 2). (ii) A control treatment with pure CaCO3 is still missing in the experimental design and this absence is not justified.

At the end, I was not convinced by the second version. I do not judge this work being of concern for the international scientific community in the topic of mushroom biology and mushroom products and it cannot be published in its present form.

Author Response

Point 1: The English language and stile need deep revision (especially for the new sentences added in the version.)

Response 1: Thank you for your valuable suggestions. We have used American Journal Experts (AJE, https://www.aje.com) to edit the English language. The editing certificate was shown as follows.

Point 2: A control treatment with pure CaCO3 is still missing in the experimental design and this absence is not justified.

Response 2: Thank you for your valuable suggestions. We preliminarily designed substrate formulas with different kinds of calcium sources to cultivate P. eryngii. CaCO3, CaSO4, CaHPO4 , (CaOH)2, and SSP were used as supplemented with base substrates (BS) for mushroom cultivation. We planned to discuss the possibility of added SSP in the cultivation of P. eryngii, then the results of CaCO3, CaSO4, CaHPO4, and Ca(OH)2 supplemented with BS for mushroom cultivation will be discussed together. So, we did not add CaCO3 treatment as a control treatment in the former version of manuscript. Considering the reviewer's suggestion, according to previous studies, we chosed the substrates added 0.5% CaCO3 (C1), which showed the best performance in the cultivation, as the control treatment named CK1 and supplemented the relevant data in the revised manuscript this time. For your information, we listed some data of all calcium source experiments at the attachment we submited.

Round 3

Reviewer 2 Report

In this second revision of the manuscript (version 3) , the two main weaknesses I pointed before have been corrected  (i) The English language and stile have been edited with American Journal Experts (ii) A control treatment with pure CaCO3 has been added. I guess the data was available but not used in this paper. It is absolutely necessary to write in the manuscript if the data came from the experiment or for another set with the same protocols and batch of raw materials. 

Following are few specific comments:

Line 15: the sentence should be ...were 14% higher after supplementation of the substrate with 2% SSP compared with....

Line 362: the heading should be 'fruiting bodies production' or mushroom cultivation.

Table 1: please check the lettters of significance. T3 and T5 should not be different

Lines 926- ... : You should comment here the differences between SSP and CACO3 in the present work.

Line1139: according to the data, 1 to 3 % are not different and = 0.5 % CaCO3. You should change the sentence.

Line 1149: You should add here that CaCO3 is necessary for improving the yield

Line 1313: Actually T5 has the better ratio proteins/fat. You focused on T2 because of the yield?. Add information in the manuscript.
You should comment hereon the absence of general trends in distribution of the components with the increasing concentration of SSP.

Line 1332: but with CaCO3, also high Ca concentration in mushroom whereas lower quantities in substrate than with SSP. You shoud comment in in the manuscruipt.

Line 1328: Ca is a macroelement and is presented above. Change the headings or delete one

Line 1344: the relative quantities of the different mineral nutrients did not vary in the same direction and with a logical trend with increasing SSP. That means that there is no rule and it canot be predicted any effect on that parameters. It should be discussed.

Line 1562: you should add that CACO3 improved the yield or BE at a same level thant 2%. The only significant difference was for the dimaters of the caps.

Line 1569: You should add "as an alternative to other Ca sources in cultivation substrates".

Author Response

Point 1: Line 15: the sentence should be ...were 14% higher after supplementation of the substrate with 2% SSP compared with....

Response 1: Thank you for your valuable suggestions. We have corrected the sentence.

Point 2: Line 362: the heading should be 'fruiting bodies production' or mushroom cultivation.

Response 2: Thank you for your valuable suggestions. We have corrected the heading.

Point 3: Table 1: please check the lettters of significance. T3 and T5 should not be different.

Response 3: Thank you for your valuable suggestions. We have checked our datas and corrected it.

Point 4: Lines 926- ... : You should comment here the differences between SSP and CaCO3 in the present work.

Response 4: Thank you for your valuable suggestions. We have discussed the differences between SSP and CaCO3 in the present work on line 247-250 in the latest revised manuscript.

Point 5: Line 1139: according to the data, 1 to 3 % are not different and = 0.5 % CaCO3. You should change the sentence.

Response 5: Thank you for your valuable suggestions. We have changed the sentence on line 304-305 in the latest revised manuscript.

Point 6: Line 1149: You should add here that CaCO3 is necessary for improving the yield

Response 6: Thank you for your valuable suggestions. We have added the necessity of CaCO3 for improving the yield on line 314 in the latest revised manuscript.

Point 7: Line 1313: Actually T5 has the better ratio proteins/fat. You focused on T2 because of the yield?. Add information in the manuscript.

Response 7: Thank you for your valuable suggestions. We have added information on line 349-350 in the latest revised manuscript.

Point 8: You should comment here on the absence of general trends in distribution of the components with the increasing concentration of SSP.

Response 8: Thank you for your valuable suggestions. We have added discussion on line 360-363 in the latest revised manuscript.

Point 9: Line 1322: but with CaCO3, also high Ca concentration in mushroom whereas lower quantities in substrate than with SSP. You shoud comment in the manuscruipt.

Response 9: Thank you for your valuable suggestions. We have added discussion on line 366-369 in the latest revised manuscript.

Point 10: Line 1328: Ca is a macroelement and is presented above. Change the headings or delete one.

Response 10: Thank you for your valuable suggestions. We have changed the headings of 3.4, 3.5, 4.4, and 4.5 in the latest revised manuscript.

Point 11: Line 1344: the relative quantities of the different mineral nutrients did not vary in the same direction and with a logical trend with increasing SSP. That means that there is no rule and it canot be predicted any effect on that parameters. It should be discussed.

Response 11: Thank you for your valuable suggestions. We have discussed it on line 391-393 in the latest revised manuscript.

Point 12: Line 1562: you should add that CaCO3 improved the yield or BE at a same level that 2%. The only significant difference was for the dimaters of the caps.

Response 12: Thank you for your valuable suggestions. We have added it on line 404-406 in the latest revised manuscript.

Point 13: Line 1569: You should add "as an alternative to other Ca sources in cultivation substrates".

Response 13: Thank you for your valuable suggestions. We have added at 412-413 in the latest revised manuscript.
